# Can the Timed and Targeted Counseling Model Improve the Quality of Maternal and Newborn Health Care? A Process Analysis in the Rural Hoima District in Uganda

**DOI:** 10.3390/ijerph18094410

**Published:** 2021-04-21

**Authors:** Geoffrey Babughirana, Sanne Gerards, Alex Mokori, Isaac Charles Baigereza, Alex Mukembo, Grace Rukanda, Stef P. J. Kremers, Jessica Gubbels

**Affiliations:** 1Department of Health Promotion, NUTRIM School of Nutrition and Translational Research in Metabolism, Maastricht University, 6229 GT Maastricht, The Netherlands; sanne.gerards@maastrichtuniversity.nl (S.G.); s.kremers@maastrichtuniversity.nl (S.P.J.K.); jessica.gubbels@maastrichtuniversity.nl (J.G.); 2Independent Researcher, 627 Ntinda, Kampala 10302, Uganda; alexkmokori@gmail.com; 3World Vision International, Hoima Plot 15B, Nakasero Road, Kampala 5319, Uganda; baigerezaisaac@gmail.com (I.C.B.); mukembo1alex@gmail.com (A.M.); 4Lutheran World Federation, Kyangwali Plot 1401 Gaba Road, Kampala 5827, Uganda; grukanda55@gmail.com

**Keywords:** maternal and newborn health, timed and targeted counselling, Village Health Teams, antenatal care

## Abstract

Each year, more than half a million women die worldwide from causes related to pregnancy and childbirth, and nearly 4 million newborns die within 28 days of birth. In Uganda, 15 women die every single day from pregnancy and childbirth-related causes, 94 babies are stillborn, and 81 newborn babies die. Cost-effective solutions for the continuum of care can be achieved through Village Health Teams to improve home care practices and health care-seeking behavior. This study aims at examining the implementation of the timed and targeted counseling (ttC) model, as well as improving maternal and newborn health care practices. We conducted a quantitative longitudinal study on pregnant mothers who were recruited on suspicion of the pregnancy and followed-up until six weeks post-delivery. The household register was the primary data source, which was collected through a secondary review of the ttC registers. All outcome and process variables were analyzed using descriptive statistics. The study enrolled 616 households from 64 villages across seven sub-counties in Hoima district with a 98.5% successful follow-up rate. Over the course of the implementation period of ttC, there was an increase of 29.6% in timely 1st antenatal care, 28.7% in essential newborn care, 25.5% in exclusive breastfeeding, and 17.5% in quality of antenatal care. All these improvements were statistically significant. The findings from this study show that the application of the ttC model through Village Health Teams has great potential to improve the quality of antenatal and newborn care and the health-seeking practices of pregnant and breastfeeding mothers in rural communities.

## 1. Background

Each year, more than half a million women die worldwide from causes related to pregnancy and childbirth, and nearly 4 million newborns die within 28 days after birth [1]. In Uganda, 15 women die every day from pregnancy and childbirth-related causes, 94 babies are stillborn, and 81 newborn babies die. This equates to 695,701 deaths each year due to complications during pregnancy, childbirth, and in the first month [2]. Therefore, the link between a mother’s health and that of her newborn is crucial for addressing maternal and infant mortality and morbidity, since the majority of preventable maternal and infant deaths occur during pregnancy, childbirth, and the immediate postpartum period [3]. Ensuring that mothers and newborns receive timely, high-quality care during these critical periods is essential. At baseline, attending four or more ANC was 55.1% [4]; therefore, this means almost half of the mothers miss goal oriented services such as screening for, preventing, and treating HIV, anemia, malaria, urinary tract infections, pre-eclampsia/eclampsia, preterm delivery, and other complications that can be life-threatening to both mother and baby [5]. Supporting women to deliver under the supervision of a skilled birth attendant and having the necessary supplies is key to preventing leading causes of maternal and newborn deaths [6].

One of the globally accepted approaches aimed at reducing maternal and infant mortality is the continuum of care approach. This approach advocates individuals having access to key services throughout the life cycle, including pregnancy, childbirth, and the postnatal period [7], to prevent maternal and newborn mortality [8]. The goal of the continuum of care approach is to provide women with the reproductive health care they need during and after pregnancy and delivery, and to provide newborns with the opportunity to grow into healthy children [9]. Emphasis is placed on a unified integrated approach for mothers and newborns and the role of facility-based care, outreach, and community-based or home care [10].

Further still, when dealing with pregnant and lactating women at the household level, an appropriate approach needs to be selected that includes the promotion of access to care through the life cycle. This approach should also focus on the involvement of male partners in household based care. This could potentially further contribute to the reduction of gender-based violence during the reproductive health period. Studies in Tanzania showed that intimate partner violence is one of the main challenges to women’s health and wellbeing during the pregnancy period. Especially young married women are vulnerable, affecting their maternal health and pregnancy outcomes. Therefore, this calls for the proper implementation of community-based interventions to support pregnant women to seek antenatal care services and to raise awareness regarding the reduction of intimate partner violence [11].

Cost-effective solutions such as working with Uganda’s community health worker cadre, called the Village Health Teams (VHTs) [12], are available. This could produce rapid improvements, but urgency and commitment are required to implement them [13]. Continuum of care through VHTs can be achieved through a combination of policies and strategies to improve home care practices, health care-seeking behavior, and health care services at the health facility throughout life, building on existing programs and packages. Home visits by VHTs together with community mobilization activities have been associated with reduced neonatal mortality and stillbirths in settings with high neonatal mortality and poor access to facility-based health care [14]. Such home visits by VHTs have been recommended by the WHO, UNICEF, and partners since 2009 [15]. In Uganda, a previous study has shown that pregnancy and newborn home visits by VHTs combined with health facility strengthening is feasible and can improve maternal and newborn care practices in a typical rural Ugandan setting [16]. The use of locally recruited, trained VHTs linked to a health facility-based supervision system to deliver home-based preventive care, was found to be acceptable and effective in rural geographical locations [17].

One of the approaches VHTs can use, the timed and targeted counselling (ttC), [18] is a family-inclusive behavior change communication approach, targeting households with pregnant women through the continuum of care. The timed and targeted counselling model encompasses a wide range of life-saving health practices through appropriately timed messages, delivered using interactive storytelling. It applies a dialogue counseling methodology, based on the assessment of current needs and practices and negotiation of progressive improvements [19]. The VHTs seek to engage parents and family decision-makers, leading to a family-inclusive and gender-transformative model of pregnancy and newborn care, in which the positive contribution of male partners is emphasized. In practical terms, the VHTs organize immunization outreaches, perform home visits and various other activities that are involved with ttC, and attend other activities in the district concerning the community. The home visits are implemented for pregnant mothers, and after birth, the children are followed up and registered. VHTs also monitor the use of certain services, for instance whether mothers come for antenatal care at the health facility and whether they have had an HIV test [4].

The behavioral change communication model of choice for this study, ttC, fits into the Healthy Places for Children [20] that underpins the critical role of engagement, common vision, and collaboration. These are seen in the involvement of the district local government: how the health providers—in this regard, the health workers and the VHT—are at the center of providing service and demand creation. The model also focuses on social inclusion at the family level where the discussions and negotiations include all influencers such as the husband, in-laws, and grandmothers.

In the current study, ttC targeted pregnancy outcomes such as referring to goal-oriented ANC, preparedness for birth, male involvement in pregnancy and newborn care, and delivery at a health facility, which are considered as contributing to positive birth outcomes [21]. This study aimed at assessing the maternal and newborn health care before and after the implementation of ttC in the rural Hoima district in Uganda. This was done by comparing the quality of care and essential services during the pregnancy and newborn period.

## 2. Methods

### 2.1. Study Design

This was a longitudinal study, with an evaluation of the implementation of ttC as the subject of follow-up and changes overtime. Quantitative measures were used to evaluate the implementation process and to investigate change over time in pregnancy and newborn care. The study team continues to work tirelessly on plans to publish a paper focusing on the effects of ttC implementation.

### 2.2. Study Area

The study was carried out in the Hoima district, located in the western part of Uganda. Hoima has 15 sub-counties and has a population of over 500,000 [22]. The area used to be predominantly inhabited by the native Banyoro, but since the discovery of oil, new economic developments such as road construction, and the establishment of factories and fishing activities, people from other locations in Uganda have been moving to the district. However, most of the households depend on subsistence agriculture and small-scale cash crops such as tobacco and coffee [23]. ttC was implemented in the Hoima district, because this district has one of the highest maternal mortality rates in Uganda [24].

### 2.3. Study Population

The study population consisted of pregnant mothers registered by VHTs from the time they suspected they were pregnant. The study followed up these participants from pregnancy until six weeks post-delivery. To be included, women had to have either an ANC card (which is used to record vital information about the pregnant woman’s service provision.) [25] or a mother–child passport [26]. After delivery, women with an ANC card would additionally be provided with a Child Health Card [27]. The VHTs [28] were considered the secondary study subjects, since they were the ones responsible for performing the ttC household visits and the counseling as well as documenting in the registers. There were no exclusion criteria for the included women, other than that the pregnant women had to have lived in the project area until at least six weeks after childbirth. Data were collected by 65 VHTs from eight sub-counties in Hoima, with a total of 616 participating women and households. These included Buseruka, Upper and Lower Kabwoya, Busisi, Kigorobya, Bugambe, Kitoba and Mparo.

### 2.4. Ethics Approval

Ethical approval and clearance were received from the Institutional Review Board (IRB) for the School of Public Health, Makerere University College of Health Sciences, the Higher Degrees, Research and Ethics Committee (HDREC) protocol 730 on February 2nd, 2020. In line with the Ugandan government regulations, the approved protocol was then submitted to the Uganda National Council for Science and Technology (UNCST) for approval. This was approved with a research registration number: HS574ES. Since this study was regarded as an anonymized desk review, there was no need for the consent of the pregnant women or mothers (Trial registration: PACTR, PACTR202002812123868. Registered on 25 February 2020—Retrospectively registered, http://www.pactr.org/ PACTR202002812123868, accessed on 25 February 2020.

### 2.5. Intervention

The ttC intervention [29] was rolled out as a behavioral change communication model for pregnancy and newborn care. For ttC to be implemented, a five-day central hands-on training was given for the VHTs by the district health team (these are district health officials qualified as public health assistants that offer supervision and guidance to the VHT), using a two-way counseling approach (participatory approach). Then, these VHTs translated the acquired knowledge to the expectant mothers and mothers with neonates, as they delivered the ttC package during their home visits.

VHTs were taught how to communicate with the pregnant women and mothers with neonates, as well as the household heads (mostly the fathers) and any other influential individuals in the family. During the training, the VHTs were equipped with a participant’s manual, and a ttC-specific household register to record participants and kick-start the timely follow-up process [30].

The ttC visits pass on a particular message at set times to trigger the household to encourage the mother to go for a given service at the health center, which is relevant at that moment in pregnancy or the neonatal period. The visits that the VHT makes are in line with the Ministry of Health’s goal-oriented ANC [24] services package and the institutional delivery framework of the road to the reduction of maternal and newborn death in Uganda [31]. For this study, the focus is on the four visits made during pregnancy, the three visits made when the child is in the newborn period for essential newborn care practices, and finally, a fourth postpartum visit at six weeks to enhance exclusive breastfeeding and routine immunization. The VHTs were convened by the sub-county assistant on a monthly basis to ensure they adhered to the prescribed household visits and registered each action in the ttC household register. The visits are further specified in Table 1.

In a ttC pilot study among 1556 mothers in Palestine, [32] CHWs targeted mothers with timely key messages and support for positive feeding and caring practices during organized home visits throughout 12 months. The pilot study showed that practices improved significantly among participating mothers: exclusive breastfeeding until six-months increased from 33% to 48.4%; the proportion of mothers who report having four or more antenatal visits increased from 58.4% to 63.6%, and the proportion of mothers who received at least two post-natal visits increased from 26.8% to 52.4%.

### 2.6. Study Instrument and Variables

The ttC household register (this is the register that the VHT uses to tick off or confirm the key services or behaviors the household has implemented at each visit) was the primary data source. Data were collected through secondary review of the existing filled out ttC household registers for mothers who had been followed from the time the VHT identified them as pregnant to at least six weeks after childbirth. These data were collected by the VHTs during their household visits. While the VHTs are doing household counseling, they record information in the household register. The study variables consist of two groups: those that were measured at baseline as well as follow-up, and those that are specific to ttC and therefore were not applicable during the baseline timeframe and thus only collected at follow-up (see Table 2 for details on the study variables).

The principle data collectors were the VHTs who collected the data during household visits at given times within the pregnancy and newborn period [28]. The VHTs were established by the Ministry of Health to empower communities to take part in the decisions that affect their health, mobilize communities for health programs, and strengthen the delivery of health services at the household level. The role of the VHTs included ensuring community participation and empowerment, which is a strategy that enables communities to take responsibility for their own health and wellbeing and to participate actively in the management of their local health services. The VHTs help to reach community participation in health and link the communities to the formal health service delivery system. This helps to bridge the current health human resource gap, especially in rural or peripheral areas, where the majority of the people live. The household counseling procedure facilitated by the VHTs on a household visit was estimated to take up to 45 minutes per session.

### 2.7. Data Management and Analysis

Access to all data was limited to the research team and secured with a strong password. The principal investigator (GB) had overall responsibility for data management over the course of the study project and monitored compliance with the protocol.

A data entry template was created in Microsoft Excel to enter the longitudinal data regarding the participants. Each household page included was given a unique code to be able to identify participants and link baseline to follow-up data. The process of unique coding included sub-county, parish, village, and the house number. Data from Excel were exported to SPSS version 20 (IBM, New York, NY, USA). Then, correctness and consistency were checked before final data files were created for analysis. Descriptive analyses (frequencies and percentages) were used for analyzing all outcome and quality of health care variables. This was done for the four outcome variables: quality of ENC, quality of ANC services at the HF, childbirth practices, and quality of ANC care at home.

Cross-tabulation using the McNemar’s Chi-square test was used to compare the percentage of the five key outcome variables before and after implementation: first ANC in first trimester, quality of ENC, quality of ANC, exclusive breastfeeding, and four or more ANC visits. *p*-values < 0.05 were considered statistically significant.

### 2.8. Relevance of the Study

Analysis in preparation for the implementation of the sustainable development goals related to maternal and newborn health [37] revealed a slow progress in the reduction of maternal and newborn mortality by the Ugandan government. Evidence from other community-based interventions, for example a previous large-scale CHW intervention, showed that CHWs were effective in identifying pregnant women in their homes early in pregnancy and before they had attended ANC. The ttC intervention is meant to fulfill some of the conditions that are necessary for CHW to improve timely ANC uptake and newborn care [38] and therefore contribute positively to healthy pregnancy outcomes and reduction of maternal mortality in Uganda.

Furthermore, ttC is in line with renewed interest in the potential contribution of strengthening community health systems, including case management of childhood illnesses (e.g., pneumonia, malaria, and neonatal sepsis), delivery of preventive interventions such as immunization, promotion of healthy behavior, and mobilization of communities [39]. Therefore, the results from this study are to be used to asses if any progress in intermediate outcomes has been made in the study district. The ttC model is a World Vision initiative that has not been evaluated before. This study will provide evidence for scale up of ttC and adaptation by the Ministry of Health, as a model of choice for community-based maternal and newborn care at the policy level. The evidence gathered can be used to encourage more expectant mothers to continue using the established government structures, while advocating for scale up to the rest of the country. If implemented accordingly, widespread dissemination to other parts of the country can improve pregnancy outcomes nationwide.

## 3. Results

### 3.1. Social Demographic Characteristics of Participants

The study enrolled 616 households with pregnant women from 64 villages, across seven sub-counties reached by 65 VHTs. In total, 53.8% of the VHTs were female. On average, each VHT was responsible for the follow up of 10 pregnant women per village. Of the 616 pregnant women enrolled, 98.5% (*n* = 607) were successfully followed up to the end of the 6 weeks after childbirth. As regards the women who dropped out, 0.8% lost the newborn, 0.3% had a miscarriage during the pregnancy period, 0.2% migrated to places outside of the project coverage, while another 0.2% were not followed up by the VHTs. Further details have been included in Appendix A.

### 3.2. Changes in Key Outcomes

In comparison to before the implementation of ttC (see Figure 1 below), there was a general improvement in pregnancy and newborn care practices at follow-up. There was an increase of 29.6% in timely 1st ANC visit, of 28.7% in quality essential newborn care, of 25.5% in exclusive breastfeeding, and of 17.5% in quality of ANC care. These improvements were all statistically significant (*p*-values < 0.001). Attendance at the fourth or further ANC visit was declined by 4.6%, which is a non-significant (*p* = 0.153) difference.

### 3.3. Quality of Care

Appendix A presents some components of quality of care at baseline and follow-up. Contributing factors to the increased quality of ANC care were an increase in iron folic supplementation of 67.0% compared to baseline and a 29.5% increase in attendance of the first ANC in the first trimester. Attendance at four or more ANC visits and intermittent presumptive treatment (IPT) during ANC showed no substantial change. In addition, a decrease in HIV testing during pregnancy by 15.0% was shown. As regards quality of care at home, there was a decrease in the percentage of women sleeping in a mosquito net by 24.7% from the baseline.

Contributing factors to appropriate childbirth practices improved by supporting the pregnant women to discuss the birth plan by 62.8%, and there was an increase of deliveries at the health facility by 10%. However, there was a decrease in the possession of a clean birthing kit of 40%. There was a steep increase in mothers practicing exclusive breastfeeding by 24.6% but also a drop in mosquito net utilization for the newborn by 11.9%.

### 3.4. Timely ANC Visits Recorded by the VHT

After implementation of ttC, 97.2% of all pregnant women attended the first ANC, with 52.2% having this service in the first trimester. Less women went for a second ANC visit (85.9%). The majority of these second visits (59.8%) occurred in the second trimester. The third ANC visit was attended by 69.3% of all pregnant women, with 40.6% having this visit in the second trimester. A 4th or further ANC visit to the HF was done by only 52.3% of all pregnant women, with 48.6% happening in the third trimester. The details of the period of each ANC visit are detailed in Figure 2 below.

### 3.5. ANC Appropriate Services and Home Practices

The provision of iron–folic supplementation to pregnant women at all ANC visits was practiced most often, with an average of 80% throughout the pregnancy visits. This showed an increasing trend from the first ANC visit through to the last before childbirth. The provision of IPT was given to 44% of pregnant women on average. As seen in Appendix A below, the majority of pregnant women took IPT during the 2nd ANC visit, and it was given to very few mothers during the 4th or further ANC visits.

HIV counseling and testing were provided to 43.3% of the pregnant women, with the majority receiving the service early on in their pregnancy. On average, 15.6% of the pregnant women were sleeping under an LLITN, with the highest reported prevalence during the second ANC visit, and the poorest reported use during the fourth or further ANC visit.

A total of 44% of the pregnant women reported having adequate rest while at home, with the majority of the women at the fourth or further ANC visit reporting resting more, compared to 18.3% at the first ANC visit. Having adequate rest showed a progressive trend with more resting when approaching the childbirth period. On average, 42.5% of pregnant women had extra meals during pregnancy, which progressively increased with pregnancy duration. Only 23.4% of the pregnant women had a hand washing facility at the latrine, with more pregnant women having the facility as childbirth approached. More details can be accessed in Appendix A.

### 3.6. Essential Newborn Care during the First Week after Childbirth

Of the three visits required to be made by the VHT during the first week of the baby’s life, the majority of the mother–newborn pairs were visited on the 7th day after birth, followed by 60.6% visited on day 3 while those visited on day 1 was 58.2%. Of the newborns that were visited on day 1, 73.4% had been given the required vaccines. The majority (83.7%) of the mothers were supported by the VHTs to exclusively breastfeed their newborns on the third day after childbirth. Appropriate newborn suckling was supported in the majority (70.7%) on the 7th day (Figure 3). Fewer mothers were supported with regard to the newborns’ cord care. VHTs focused on cord care progressively, with fewer mothers being supported on the 1st day and more mothers being supported on the 7th day.

### 3.7. Newborn Care at Six Weeks after Childbirth

Appendix A shows that 84.8% of the newborns were visited by the VHTs at 6 weeks after childbirth. Of the visited newborns, 79.6% were sleeping under an insecticide-treated mosquito net, 61.4% still received exclusive breastfeeding, and 87.2% had been vaccinated for polio1 and DPT1. Furthermore, 84.4% of newborns had their child health card plotted at immunization, whereas only 43.3% of the mothers were prepared to use family planning, and 29.9% had been taught how to prepare ORS for diarrhea care. Appendix A provides more details of these figures.

### 3.8. Discussion

The current study examined the improvement of maternal and newborn health care and practices before and during implementation of ttC. Evaluations included the quality of care and the implementation of essential services during the pregnancy and newborn period. As observed from the results, there were noticeable and significant improvements in pregnancy and childbirth care after the implementation of ttC by the VHTs, especially in timely first ANC attendance, quality of ANC services, quality of essential newborn care, and exclusive breastfeeding. This achievement is attributed to the role of the VHTs, where community mobilization and household follow-up was done promptly and routinely to ensure pregnant mothers visit the HF to get appropriate care. The study results are consistent with evidence from Rwanda, where community health workers were deployed to improve maternal health, undertake health promotion activities, and connect people, especially those living in rural areas, with the health facilities as necessary [40]. The percentage of births assisted by skilled birth attendants in this Rwandan study increased from 26.7% to 69.0%, which is in line with the increase in deliveries in an HF in the current study [40]. In addition, the rate of contraceptive use increased from 4% to 45.1% in the study from Rwanda [40], which is in line with the 43.3% that indicated having decided on a family planning method after birth following the implementation of ttC in the current study.

However, the present study did show that after ttC implementation, attendance at the fourth and further ANC visits did not show an increase. This is surprising, since the study did show an improvement of 29.6% in early attendance of ANC, which is attributed to the role the VHTs played in ensuring pregnant women are supported at the household level. The WHO guidelines encourage pregnant women to have eight ANC contacts with the health system during each pregnancy, including the more familiar model of clinic-based ANC visits, as well as ANC care and/or counselling sessions for pregnant women at the household and community levels [41]. Attendance at the fourth and further ANC visits is often lowest among poor women with little education and who are living alone [42]. Accessibility-related factors potentially influencing low ANC attendance include long distance from facilities providing services, mode of transport, working hours, booking appointments, length of stay at the health facility, and the quality of care given in the previous visits [43].

The current study showed that recommended home-based practices during pregnancy were adhered to more often as the pregnancy approached childbirth. This can be seen for both the trends of the pregnant woman having adequate rest as well as for the pregnant woman having an extra meal during pregnancy. An extra meal provides the nutrients for the needs of the growing baby [42]. For example, extra calcium helps make and keep bones and teeth stronger than before the woman got pregnant [44]. The generally poor link between good nutrition during pregnancy and pregnancy outcomes can be seen in Uganda, where 10% of all newborns have a low birth weight at less than 2.5 kilograms [45].

Results from the study revealed that 30% of all newborns were not visited by the VHT at the household level by the close of the first week after childbirth. This is a missed opportunity for these newborns, since three-quarters of all neonatal deaths globally occur during the first week of life, contributing to the critical gaps in the continuum of care [46]. It is important to note that even if birth occurs at a health facility, in many settings, mothers and newborns are discharged within a few hours and have no further contact with a health provider until the 6-week postpartum and immunization visit [43]. This stresses the importance of the VHT visiting during the first week after childbirth [46].

This study had several strengths and limitations that need to be considered. The study provided unique insights in an understudied population. However, the study relied on secondary data, which meant that our scope was limited to the data that were previously gathered. This included lacking baseline data for a number of outcomes. Moreover, the data were collected by the VHTs, who are semi-literate in Uganda. Therefore, this meant that our data depend on what the VHTs collected according to their ability. During data analysis, variables that were transferred from the ANC card or CHC to the ttC household register by the VHT presented substantial missing data due to the lack of the ability to properly read and write, and this could have affected the percentages presented in the study. In addition, the current data did not allow for advanced statistical analyses. Therefore, this calls for a household-focused study, collecting data directly from the mother–baby pairs who have gone through ttC. Furthermore, the lack of a control group in the study brings about uncertainty of whether the changes can be attributed solely to ttC, as other concurrent factors in the region might have played a role.

## 4. Conclusions

Results from the study show an overall substantial improvement in the quality of care during the pregnancy and newborn period. Therefore, this indicates that ttC delivered through VHTs can play an integral role in ensuring the continuum of care for pregnancy and newborn care linking the community interventions to the health facilities. However, there is a need for the health system to eliminate health facility barriers that are still creating obstacles to making this process a reality.

## Figures and Tables

**Figure 1 ijerph-18-04410-f001:**
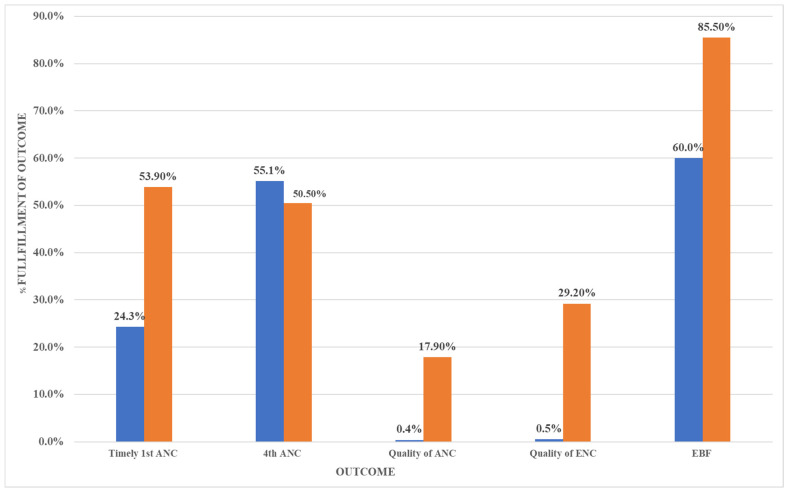
Change in key outcome variables. 
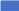
 = baseline values; 
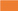
 = post-implementation values; ANC = Antenatal care; ENC = Essential Newborn care; EBF = Exclusive Breastfeeding.

**Figure 2 ijerph-18-04410-f002:**
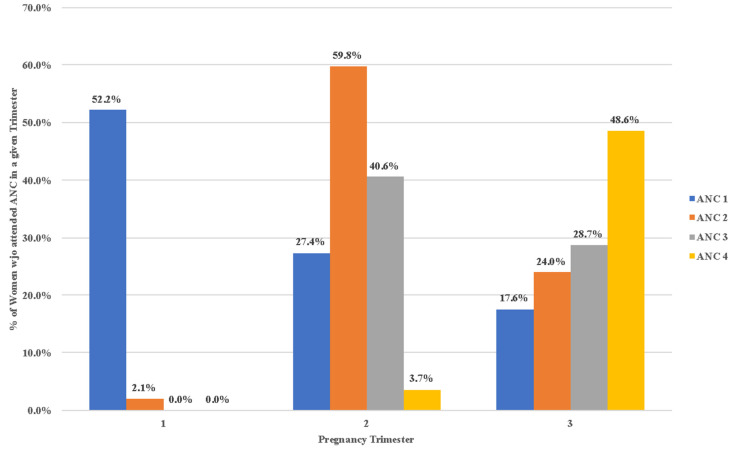
ANC visits per Trimester. ANC = Antenatal care.

**Figure 3 ijerph-18-04410-f003:**
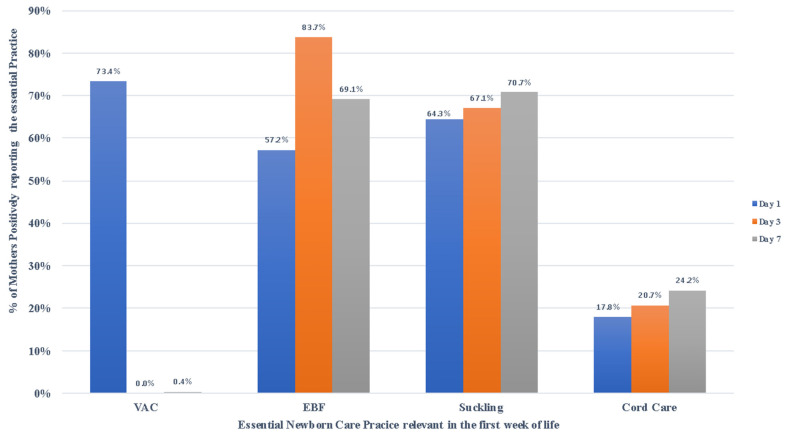
Essential newborn care during the first week. VAC = Newborns reported being vaccinated immediately after childbirth; EBF = Support to exclusively breastfeed; Suckling = Support to appropriate positioning and attachment of the newborn on the breast during feeding; Cord care = Support to practice appropriate cord care.

**Table 1 ijerph-18-04410-t001:** ttC visits by VHTs to mothers during pregnancy or after childbirth [33].

Visit	Timing	Services Delivered by the VHTs
1	Before or at 3 months of pregnancy	Focuses on suspected pregnant mothers registered as ttC candidates. They are counseled on how to care for themselves during pregnancy, a number of danger signs during pregnancy are explained, and the family is encouraged to notify the VHT in the case of any of these danger signs. Families are counseled on what to do during pregnancy and when giving birth. Women are advised to start ANC before or at three months pregnancy.
2	4 months of pregnancy	This visit is to affirm that the pregnant mother went for the first ANC visit and to encourage them to go for the second ANC visit. Pregnant women are educated on the advantages of exclusive breastfeeding of children until 6 months and continuous breastfeeding until they reach the age of 2. In addition, advantages of giving birth in a health facility and hand washing are highlighted.
3	6 months of pregnancy	Discussion of the birth plan, timing, and place of the birth, exploring potential challenges faced by expectant mothers in reaching the health facility and ways of overcoming these challenges. Expectant mothers are given information on the different family planning methods and where to go for services, and they are encouraged to utilize one of the methods after delivery.
4	8 months of pregnancy	The mother and family are educated about the early signs of true labor, danger signs during labor, and appropriate action to take in the case of these danger signs. The mother is also encouraged to deliver at a health facility and have emergency money available. In addition, the VHTs check whether the mother has a clean birthing kit available (the “Mama Kit”).
5–7	The newborn period (Days 1, 3, and 7)	This visit is done three times in the first week after birth, on days 1, 3, and 7. The VHT ensures that exclusive breastfeeding and essential maternal and newborn care are understood and practiced. Mothers are educated on the need to seek health care if the neonate develops fever and/or a cough, and the need to go for routine growth monitoring and immunization.
8	When the baby is 6 weeks old	Malaria prevention is discussed, as well as other illnesses, danger signs in the baby, immunization for the baby, and hygiene practices. The VHT checks whether the child’s growth card has been plotted, gives support for exclusive breastfeeding, checks if the child has had its HIV confirmatory test as part of the prevention of mother-to-child transition of the HIV program, and whether the mother has thought about family planning methods.

Note: ANC = antenatal care, ttC = timed and targeted counseling, VHT = village health team, HIV = Human Immunodeficiency Virus.

**Table 2 ijerph-18-04410-t002:** Study variables.

Variable	Description	Assessed at Baseline (Y/N) ^a^
Key outcome variables		
Quality of ANC services	The pregnant woman is able to visit a health facility for ANC and receive the eight key services in the goal-oriented ANC recommendations.	Y
Quality of ANC care at home	The pregnant woman performs the key four practices at home appropriate for the pregnancy period [34]. These include: woman having adequate rest, extra meal, sleeping LLTIN, and having a hand-washing facility at the home.	N
Appropriate childbirth practices	The pregnant woman is supported to practice the four key lifesaving actions prior to or during the time of birth [35]. These include plans to deliver at a health facility, discussion of a birth plan, delivery at the health facility, and provision of a clean birthing kit.	N
Timely ANC attendance	The pregnant woman attends ANC visits in a timely manner as recommended by the WHO and the Ugandan goal-oriented ANC [36].	Y
Quality of essential newborn care	Mothers performing the six key essential newborn care practices both at the health facility and at home within the newborn period of 28 days after birth. These include mother was visited by the VHT during the 1st week of birth, baby sleeping under LLTIN, practicing exclusive breastfeeding, suckling well, birth weight of the baby recorded in the CHC, and the baby has been vaccinated.	Y
Exclusive breastfeeding	Mother was able to sustain exclusive breastfeeding to the baby up to 6 weeks.	Y
Quality of care variables		
Quality of ANC services		
Woman given folic acid during ANC	Pregnant woman is given iron–folic acid tablets at all ANC visits to the health facility.	Y
Woman given Fansidar during ANC	Pregnant woman taking a directly observed dose of Fansidar at the health facility during ANC.	Y
Woman tested for HIV and results given	Pregnant woman does a routine HIV test at the health facility during the ANC visit.	Y
Woman given deworming during pregnancy	Pregnant woman is given a dewormer at the health facility during the ANC visit.	N
Woman vaccinated for Tetanus Toxoid (TT) once	The pregnant woman is given at least one dose of TT vaccine during the pregnancy period by the health facility.	N
ANC first trimester	Pregnant woman visiting the health facility for her first ANC visit within the first 12 weeks of pregnancy.	Y
4th ANC or more	Pregnant women has attended the health facility for four or more ANC visits.	Y
Woman tested for syphilis	Pregnant woman takes a syphilis [26] test at least once during the pregnancy period during ANC and receives the results of the test.	N
Quality of ANC care at home		
Woman having adequate rest	The pregnant woman is deliberately taking rest from laborious work during the day, and this is due to the pregnancy condition.	N
Woman having an extra meal	The pregnant woman consumes an extra meal on top of her usual meal frequency when not pregnant.	N
Woman sleeping in a long-lasting insecticide treated net (LLITN)	The pregnant woman slept under an insecticide-treated bed net the night before the visit made by the VHT.	Y
Hand washing facility at the home	There is a visible hand washing facility on the walkway to the latrine of the household.	N
Appropriate childbirth practices		
Woman plans to deliver at health facility	The pregnant woman is planning to deliver at the nearest health facility.	N
Woman discussed a birth plan	During the 3rd trimester, the pregnant women has discussed a birth plan with the VHT on one of the visits.	Y
Delivery at the health facility	The pregnant woman actually delivered at the health facility.	Y
Woman has a clean birthing kit	The pregnant woman is given a clean birthing kit by the health facility.	Y
Quality of Essential Newborn Care (ENC)		
Mother visited during the 1st week	The VHT performs at least one visit to the household during the 1st newborn week.	N
Baby sleeping under an LLITN	During the newborn period visits, the newborn is sleeping under an insecticide-treated bed net.	Y
Practicing exclusive breastfeeding	For the 24 hours previous to the household visit, the mother has not given anything to the newborn except for breast milk apart from medication prescribed by medical personnel to the baby.	Y
Baby suckling well	The VHT is observing breastfeeding and notices that the baby is suckling well.	N
Birth weight record	At discharge from the health facility, the newborn is weighed, and the weight is recorded on the child health card.	N
Vaccination	The newborn is given the vaccines appropriate for that time period.	N
Infant visited at 6 weeks		
Visitation at 6 weeks	The VHT checked on the 6-week vaccinations, whether the baby was taken for growth monitoring and promotion, and whether the mother knew how to prepare oral rehydration salts in the case of a diarrheal episode. The VHT also checked whether the mother had a discussion at the HF about family planning choices.	N

Notes: ^a^—Y = Yes the variable was measured at baseline, N = No, the variable was not measured at baseline. ANC = Antenatal care; HF = Health facility; VHT = Village Health Team member; IPT = Intermittent presumptive treatment; TT = Tetanus Toxoid vaccine; LLITN = Long-lasting insecticide-treated net.

## Data Availability

The raw data will be made available upon reasonable request from the corresponding author.

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
