# Peer review of "Can the Timed and Targeted Counseling Model Improve the Quality of Maternal and Newborn Health Care? A Process Analysis in the Rural Hoima District in Uganda"

_ijerph, 2021, doi:10.3390/ijerph18094410_

Round 1
Reviewer 1 Report
The study tries to evaluate the effect of an educational effort on adherence to antenatal care and ggod perinatal practices in a population of Uganda. As reported by the authors “The study provided unique insights in an understudied population” even if on secondary data, lacking baseline data for a number of outcomes, and with data collected by semi-literate researchers.
An important methodological issue/limit is how can the authors demonstrate that the improvement observed in antenatal care and some outcomes was effectively related the “ttC” new model introduced and not to other factors. When was the beginning of the ttC intervention? How many women were enrolled before and how many women after the intervention?
More details should be reported on the characteristics of the team giving the intervention/care, eventual payment needed for the interventions offered, time dedicated to the mother/couple education
More data on “before and after” characteristics of population and variable assessed should be reported if available, and a minimal set of data on the outcome of pregnancy could be useful
Author Response
Dear Reviewer,
Thank You very much for your valueable support reviewing this paper. Attached are the response

Reviewer 2 Report
This manuscript is well written but needs correction of some important points before publication. Like in introduction need to add information from the below recent paper-
https://www.nepjol.info/index.php/JCMSN/article/view/17802
Add more additional information about the research context and the reasons for undertaking this research and how this research will make an impact for the policymakers.
The study is conducted in a specific place hence the place name should be included in the title of the research. Why this place was particularly selected?
Please explain the adopted strategy for validation of the study questionnaire. Which test was used for internal validation? Mention whether there was a pilot study?How many respondents were included in pilot run? When was the data collection period? Who was responsible for data collection?
Why the p value findings are not presented in a tabular format?
There is no advanced analysis is performed. The major drawback of the study is the lack of advanced analysis.
Author Response
Dear Reviewer,
Please find attached the responses on the comments and concerns. We are so humbled by Your input to make our paper better.

Round 2
Reviewer 2 Report
The authors' have done extensive changes as per the review comments.
Author Response
Dear Reviwer
We have addressed the major comments and attached. One paragraph has been added in the background.
Thank You for Your support
